# Prevalence and Risk of Sarcopenia in Patients with Chronic Pancreatitis: Systematic Review and Meta-Analysis

**DOI:** 10.3390/nu17050870

**Published:** 2025-02-28

**Authors:** Alsu R. Khurmatullina, Dmitrii N. Andreev, Igor V. Maev, Yury A. Kucheryavyy, Petr A. Beliy, Aida R. Dzhafarova, Valeriya V. Cherenkova, Filipp S. Sokolov

**Affiliations:** 1Department of Propaedeutics of Internal Diseases, Gastroenterology and Hepatology, I.M. Sechenov First Moscow State Medical University (Sechenov University), 19435 Moscow, Russia; 2Department of Internal Disease Propaedeutics and Gastroenterology, Russian University of Medicine, 127473 Moscow, Russia; 3Ilyinskaya Hospital, 143421 Krasnogorsk, Russia; 4A.S. Loginov Moscow Clinical Scientific Centre, 111123 Moscow, Russia; valeriya_star@mail.ru

**Keywords:** chronic pancreatitis, sarcopenia, meta-analysis, systematic review

## Abstract

Background/Objectives: Sarcopenia is a condition marked by a continuous decline in skeletal muscle strength and volume, often leading to significant health complications. According to several articles, sarcopenia is highly prevalent in chronic pancreatitis (CP) due to exocrine pancreatic insufficiency. The aim of this meta-analysis was to determine the pooled prevalence and risk of sarcopenia among CP patients. Methods: The search process adhered to the PRISMA 2020 guidelines and was registered in PROSPERO under the identification number CRD42025637059. The search was conducted in the following databases: MEDLINE/PubMed, EMBASE, Cochrane, Google Scholar, and the Russian Science Citation Index (RSCI). It covered studies published between 1 January 1985 and 20 December 2024. Only studies published in English or Russian with detailed comprehensive statistics and adult CP were included. Studies with specific patient populations affecting data objectivity were excluded. Sensitivity analyses were conducted (first, only studies with more than 50 CP patients were considered. Second, the analysis was restricted to full articles, excluding abstracts from conferences). Results: In total, 16 studies with 1556 participants (1398 CP patients and 158 controls) met the criteria. The pooled prevalence of sarcopenia was 39.117% (95% CI: 28.891–49.852) in CP patients and 7.745% (95% CI: 2.154–42.622) in the control group. An association was found between sarcopenia and CP using the fixed-effects model when compared to the control group (RR = 2.194, 95% CI: 1.502–3.203). Conclusions: Sarcopenia is underdiagnosed in CP patients despite its significant clinical impact. Management strategies, including pancreatic enzyme replacement therapy, nutritional support, and resistance training show potential in the treatment of this state. Further research is needed to establish standardized diagnostic criteria and unified treatment approaches. Early detection and comprehensive care are essential to improving outcomes in CP patients with sarcopenia.

## 1. Introduction

Sarcopenia is a syndrome marked by a progressive decline in skeletal muscle mass loss, leading to a deterioration in strength [1]. Although this condition is considered a feature of aging, it can also result from various diseases, including cancer, nonalcoholic fatty liver disease, chronic pancreatitis (CP) and many others [2,3,4]. According to a recent report in 2021 including data from 151 studies, the global prevalence of sarcopenia in adults is up to 27% (95% CI: 23.0–31.0%), using the overall muscle mass definition; however, the prevalence may vary depending on the diagnostic criteria and populations included in studies [5].

One of the most important features of sarcopenia is its gradual onset, which can significantly impact patients’ quality of life [6]. Sarcopenia is defined clinically based on reductions in muscle (handgrip) strength, muscle mass, and gait speed [7,8,9]. In some cases, sarcopenia may manifest in atypical ways, particularly in the early stages, such as increased fatigue and difficulty performing routine activities [10]. The European Working Group on Sarcopenia in Older People 2 (EWGSOP2) has proposed revised diagnostic criteria that emphasize low muscle strength as a primary indicator of probable sarcopenia, with confirmation requiring evidence of low muscle mass [11].

Beyond muscle deterioration, sarcopenia contributes to metabolic dysfunction, including impaired insulin sensitivity, persistent inflammation, and weakened bone density [12]. Sarcopenia increases expenses of healthcare systems as patients with this condition are in need of long-term care [13]. Currently, sarcopenia is becoming increasingly common, especially in countries with an aging population [14]. This highlights the need for effective prevention and management strategies.

Sarcopenia is a significant complication for individuals with chronic pancreatitis (CP), an inflammatory disease of the pancreas, leading to its dysfunction [15]. The long-lasting nature of CP frequently results in malnutrition (due to exocrine insufficiency), which leads to the deteriorated absorption of essential nutrients, especially proteins and fat-soluble vitamins [16]. Furthermore, nutritional deficiency is complemented by systemic inflammation associated with CP, which promotes catabolic processes that lead to muscle deterioration [17].

A multidisciplinary approach is essential for the effective treatment of sarcopenia in CP. Pancreatic enzyme replacement therapy (PERT) is a crucial element in restoring nutrient absorption and correcting nutritional deficiencies [18]. Nutritional interventions such as a high-protein diet and taking control of essential amino acids can enhance muscle protein synthesis [19]. Furthermore, special training programs can assist in maintaining muscle mass and improving physical functionality [20]. It is essential to identify sarcopenia in CP patients at an early stage through routine assessments of muscle mass and strength. This allows interventions to be implemented and leads to an improvement in overall prognosis.

Unfortunately, there are currently no globally unified criteria for sarcopenia that can be implemented in every country. This creates challenges in objectively determining a generalized model for the global prevalence of this pathological condition in various patient groups, including those with CP. Previous systematic reviews and meta-analyses have indicated that sarcopenia is highly prevalent in CP patients, exceeding 30% in the analyzed studies [21,22]. However, these studies had limitations, including small sample sizes (Bundred J et al. [21] included only nine studies in the article where 977 patients were analyzed; Maev IV et al. [22] included seven studies in the article where 609 patients were analyzed), and lack of control groups. Additionally, previous meta-analyses focused primarily on prevalence, without assessing the relative risk (RR) of sarcopenia in CP patients compared to healthy individuals. Understanding prevalence of sarcopenia in CP patients justifies the need for routine screening programs, increased nutritional support, and rehabilitation services, as we know sarcopenia is associated with poor clinical outcomes (e.g., increased hospitalizations, complications, and mortality), a higher RR highlights CP patients as a high-risk group needing more intensive management.

To address these gaps, this meta-analysis synthesizes data from a broader set of studies to provide a more comprehensive estimate of the prevalence of sarcopenia in CP patients and determine the associated risk compared to healthy controls. By incorporating newly published data and applying rigorous inclusion criteria, this study aims to strengthen the evidence base and highlight the clinical significance of sarcopenia in CP.

Therefore, the objective of our meta-analysis is to determine the prevalence and risk of sarcopenia in CP patients compared to healthy controls by synthesizing data from a broader set of studies.

## 2. Materials and Methods

### 2.1. Study Sources and Search

The search process adhered to the PRISMA 2020 guidelines [23], with prior registration in the PROSPERO database (CRD42025637059) to ensure methodological transparency. The search was conducted in multiple databases, including MEDLINE/PubMed, EMBASE, Cochrane, the Russian Science Citation Index (RSCI), and Google Scholar, covering studies published between 1 January 1985 and 20 December 2024. Study selection was based on an analysis of titles and abstracts within these databases. The search terms for the MEDLINE/PubMed database were “(“Sarcopenia”[MeSH] OR sarcopenia OR muscle loss OR muscle wasting OR muscle atrophy OR skeletal muscle depletion OR cachexia OR frailty) AND (“Chronic Pancreatitis”[MeSH] OR chronic pancreatitis OR pancreatic exocrine insufficiency OR PEI OR pancreatic disease) AND (“Body Composition”[MeSH] OR body composition OR “Muscle, Skeletal”[MeSH] OR skeletal muscle OR “Sarcopenia”[MeSH] OR “Exercise”[MeSH] OR exercise OR handgrip strength OR grip strength OR gait speed OR appendicular lean mass OR ALM OR skeletal muscle index OR SMI OR DXA OR dual-energy X-ray absorptiometry OR bioelectrical impedance OR BIA OR CT OR MRI)”, other search strategies can be found in the Appendix A.

### 2.2. Study Selection

The studies were included in this meta-analysis if they met the following criteria: published in English or Russian, provided detailed descriptive statistics (descriptive statistics were defined as the analytical process used to extract and summarize the number of sarcopenic and non-sarcopenic patients, providing a clear quantitative representation of their distribution within the study population), focused on adult patients with CP, described clearly the methodological approach for sarcopenia and CP diagnostic, and did not involve patient populations with advanced malignancies (e.g., pancreatic cancer, gastrointestinal malignancies, and metastatic disease) due to the high prevalence of cancer-related cachexia—chronic systemic inflammatory diseases (e.g., rheumatoid arthritis, systemic lupus erythematosus, and inflammatory bowel disease), and uncontrolled endocrine disorders (e.g., Cushing’s syndrome, hyperthyroidism, and severe insulin resistance syndromes)—known to affect muscle metabolism that can independently contribute to muscle loss. Studies that did not differentiate CP patients from other pancreatic conditions (e.g., acute pancreatitis, pancreatic neoplasms) were also excluded. In cases where two or more publications presented overlapping data, only one was included in the final analysis. The quality of the included studies was evaluated using the Newcastle–Ottawa Scale (NOS).

### 2.3. Data Extraction

Two independent researchers (A.R.K. and D.N.A.) obtained information using uniform templates. Extracted data included publication year, country of study, approach used for CP identification, sarcopenia diagnostic standards, number of patients with CP, quantity of controls if applicable, and quantity of sarcopenia cases in both groups. Using the Rayyan tool, two independent reviewers (D.N.A. and A.R.K.) screened the titles and abstracts of all retrieved records, followed by a full-text review of potentially relevant studies. Key study characteristics and outcome data were independently extracted by two authors (A.R.D. and I.V.M.) using a standardized data extraction form. Whenever possible, study protocols were identified, and corresponding authors were contacted for clarifications or to obtain missing data from published reports. Any disagreements were resolved through discussion or by consulting a third reviewer (Yu.A.K.). In cases where study relevance was uncertain, additional reviewers (V.V.Ch., P.A.B., and F.S.S.) were involved in the decision-making process. Inter-rater reliability was assessed using Cohen’s kappa statistic (κ) to evaluate agreement on study inclusion, data extraction, and quality assessment. The following thresholds were used:-κ < 0.20: poor agreement;-κ = 0.21–0.40: fair agreement;-κ = 0.41–0.60: moderate agreement;-κ = 0.61–0.80: substantial agreement;-κ = 0.81–1.00: almost perfect agreement.

For this meta-analysis, the kappa statistic for study inclusion was 0.75, and for data extraction, it was 0.92, indicating almost perfect agreement.

### 2.4. Statistical Analysis

Statistical analysis was performed using MedCalc Statistical Software Program 23.0.6 (Ostend, Belgium) on Microsoft Windows 11 (Microsoft Corporation, Redmond, WA, USA). Overall frequency estimates for sarcopenia prevalence in CP patients and control groups were reported with 95% confidence intervals (CIs). Study heterogeneity was evaluated using Cochrane’s Q test and statistics, with significant heterogeneity defined as *p* < 0.05 and >75%. To address substantial heterogeneity a subgroup analysis was conducted: firstly, we evaluated the proportion of sarcopenic patients in the study group with NOS score lower than 7; secondly, we evaluated the proportion of sarcopenic patients in the study group with NOS score more than 7. Potential publication bias was assessed using a funnel plot, the Begg–Mazumdar correlation test, and Egger’s test.

## 3. Results

### 3.1. Search Results

The electronic database search retrieved 770 studies for initial screening. Of these, 492 studies were excluded because they did not meet the inclusion criteria (158 reviews and systematic reviews, 25 case reports, 153 irrelevant studies, and 156 duplicates). The remaining 278 studies underwent detailed evaluation, leading to the exclusion of a further 262 studies (170 studies presented wrong population; 98 studies were on an inappropriate topic; 22 studies had an unapplicable outcome) (Figure 1). Ultimately, 16 original studies met eligibility criteria and were incorporated into the final meta-analysis (Table 1) [24,25,26,27,28,29,30,31,32,33,34,35,36,37,38,39].

A total of 16 studies were included in the analysis, covering a total of 1556 individuals, consisting of 1398 patients diagnosed with CP and 158 healthy controls. The studies were performed in Russia (*n* = 3) [24,25,33], Ireland (*n* = 1) [26], India (*n* = 1) [38], France (*n* = 1) [27], Lithuania (*n* = 2) [28,30], Denmark (*n* = 1) [29], the United States of America (*n* = 2) [31,32], Great Britain (*n* = 2) [34,37], Germany (*n* = 1) [35], Finland (*n* = 1) [36], and Japan (*n* = 1) [39]. Control groups were included in three of these research articles [33,35,37].

In the majority of studies, sarcopenia was diagnosed using validated diagnostic techniques, specifically by calculating the musculoskeletal index at the third lumbar muscle level (*n* = 11) [24,25,26,27,28,30,32,34,35,38,39]. The evaluation of methodological quality using the NOS identified six studies with a low risk of bias, scoring 7 or more [27,30,31,35,36,37].

### 3.2. Prevalence of Sarcopenia in CP Patients

The overall prevalence of sarcopenia in CP patients and controls was 39.117% (95 CI: 28.891–49.852; Figure 2) and 7.745% (95 CI: 2.154–42.622), respectively. The association between the two remained significant when focusing only on studies that clearly used validated methods for diagnosing both CP and sarcopenia, with a prevalence of 45.955% (95% CI: 31.727–60.528; $I^{2}$ = 93.73%). Due to considerable heterogeneity among groups ($I_{CP}^{2}$ = 93.71%, $I_{control}^{2}$ = 96.60%; *p* < 0.0001), a random-effects model was applied. To investigate possible publication bias, a funnel-shaped plot was created and the Begg–Mazumdar and Egger’s statistical tests were conducted. In this analysis, the intercept of Egger’s regression is 4.7935, with a 95% CI of −0.7495 to 10.3365. The *p*-value of 0.0848 suggests that there is no strong statistical evidence of publication bias, as it does not reach the conventional significance threshold of *p* < 0.05. However, the relatively wide confidence interval indicates some uncertainty, meaning that while substantial publication bias is unlikely, it cannot be entirely ruled out.

A visual examination of the funnel plot (Figure 3) did not indicate notable asymmetry. Additionally, the Begg–Mazumdar test (*p* > 0.05) and Egger’s test (*p* > 0.05) provided no evidence of significant publication bias.

### 3.3. Risk of Sarcopenia in CP Patients

A significant association was found between sarcopenia and CP when compared to the control group. Based on the fixed-effects model, the relative risk (RR) was estimated as 2.194 (95% CI: 1.502–3.203; $I^{2}$ = 43.26%, 95% CI: 0.00–82.96%).

### 3.4. Subgroup Analysis

The subgroup analysis based on the NOS scores effectively addresses the heterogeneity of the included studies by stratifying them according to methodological quality. By separately analyzing studies with NOS scores of 7 or higher and those with scores below 7, the impact of study quality on the pooled prevalence of sarcopenia in CP patients can be assessed. The results indicate that studies with higher NOS scores reported a pooled prevalence of 31.680% (95% CI: 12.793–54.443) with high heterogeneity (*I*^2^ = 94.99%; 95% CI: 91.52–97.04%), whereas studies with lower NOS scores reported a higher pooled prevalence of 44.333% (95% CI: 32.285–56.729), also with high heterogeneity (*I*^2^ = 92.88%; 95% CI: 88.96–95.42%). The subgroup analysis is still valid despite the high heterogeneity because it demonstrates that the heterogeneity is present in both high-quality and lower-quality studies, suggesting that factors beyond study quality contribute to the variability in prevalence estimates. Since the *I*^2^ values remain high in both subgroups (94.99% for NOS ≥ 7 and 92.88% for NOS < 7), this indicates that heterogeneity is not solely driven by study quality but likely by other factors such as differences in study populations, diagnostic criteria for sarcopenia, or variations in CP severity.

### 3.5. Sensitivity Analysis

In the sensitivity analysis, two specific inclusion criteria were applied. First, only studies with more than 50 CP patients were considered [28,29,30,32,33,34,35,37,38,39], ensuring a sample size large enough to provide meaningful insights: the overall proportion of sarcopenia in CP patients was 35.350% (95 CI: 22.914–48.889). Second, the analysis was restricted to full articles, excluding abstracts from conferences [25,27,28,29,30,31,32,33,34,35,36,37,39], to ensure a more rigorous and comprehensive review: the overall prevalence of sarcopenia in CP patients was 29.862% (95 CI: 24.769–47.294).

## 4. Discussion

This meta-analysis represents the most comprehensive assessment to date of sarcopenia prevalence in CP patients. The results indicate that CP patients have a significantly higher risk of developing sarcopenia, with an overall prevalence of 39.117% (95% CI: 28.891–49.852%), compared to only 7.745% (95% CI: 2.154–42.622%) in healthy controls. The estimated RR of sarcopenia in CP patients was 2.194 (95% CI: 1.502–3.203), confirming a strong association between CP and muscle loss.

CP is a chronic inflammatory condition of the pancreas characterized by structural changes in the pancreas, leading to its exocrine insufficiency, which increases the likelihood of malnutrition and sarcopenia [40]. Sarcopenia, which is defined by the progressive loss of skeletal muscle mass and strength, is a common complication in CP patients. In 2020, a review of the literature estimated that approximately 16–62% of CP patients depending on the diagnostic criteria and study population suffer from sarcopenia [41]. Sarcopenia in CP patients often presents alongside malnutrition and systemic inflammation, exacerbating morbidity and impacting life quality [42].

The development of sarcopenia in CP is multifaceted. A primary contributor is pancreatic exocrine insufficiency (PEI), which results in nutrient malabsorption, particularly of proteins, fats and, as a consequence, vitamins A, D, E, and K [43]. In addition to macronutrient deficiencies, micronutrient deficiencies are prevalent in CP patients, contributing to muscle weakness and fatigue. Low levels of B vitamins (B1, B6, B12, and folate) impair energy metabolism and neuromuscular function [44], while deficiencies in magnesium and zinc can negatively affect muscle contraction and repair [45]. Iron deficiency is also common, often leading to anemia and reduced oxygen delivery to muscles, further impairing physical performance [46].

Moreover, reduced appetite and dietary intake due to chronic pain, nausea, and early satiety contribute to insufficient caloric consumption, exacerbating weight loss and muscle depletion [47]. Many CP patients also experience alcohol-induced nutritional deficiencies, as chronic alcohol consumption, a major risk factor for CP, further impairs nutrient absorption and utilization [48].

Endocrine insufficiency, commonly emerging as diabetes mellitus, contributes further to muscle loss through hyperglycemia and insulin resistance [49]. Additionally, chronic systemic inflammation accompanying CP causes muscle degradation through the activation of pro-inflammatory cytokines such as TNF-α and IL-6 [50]. Importantly, the diagnosis of CP is often delayed in asymptomatic or pain-free cases, with up to 10% of patients presenting without abdominal pain. This diagnostic delay often coincides with the onset of exocrine pancreatic insufficiency (EPI), a critical factor contributing to sarcopenia [51].

An early systematic review conducted in 2022 included data from nine studies involving 977 patients and reported a sarcopenia prevalence of 32.3% in CP patients (95% CI: 22.9–42.6%). Sarcopenia was associated with worse long-term outcomes, including a one-year mortality rate of 16% in sarcopenic patients compared to 3% in patients without sarcopenia (HR: 6.69, 95% CI: 1.79–24.9; *p* < 0.001) [21]. These findings also align with our previous meta-analysis: 42.09% (95% CI: 27.845–57.055%) of CP patients developed sarcopenia [22]. Unfortunately, these studies did not consider control groups, and few studies were included. Therefore, the overall statistical power and reliability of the conclusions and the studies’ ability to identify true effects of the intervention were limited. Sarcopenia remains an underdiagnosed complication of CP despite its significant impact on patient outcomes. Studies indicate that it increases the hospitalization rates, prolongs the length of stay at the hospital, and decreases survival in CP patients [29]. These important findings underscore the need for timely identification and management of sarcopenia as part of comprehensive CP care.

Management strategies include nutritional interventions (adequate calorie and protein intake and vitamin D) and resistance training, primarily PERT as the primary treatment for EPI, to elevate nutrient absorption [52]. Targeting energy alone leads to further muscle loss, so in order to gain weight, sarcopenic patients need to consume an additional 200–750 kcal per day [53]. European Society for Clinical Nutrition and Metabolism (ESPEN) guidelines recommend protein consumption of 1.2–1.5 protein/kg body weight daily for sarcopenic individuals with acute or chronic diseases [54].

Research suggests that some individuals with pancreatic disorders may require a substantial vitamin D dose—up to 20,000 IU per day—because of the decreased intestinal absorption caused by EPI [55]. Additionally, a meta-analysis conducted by a group of researchers concluded that resistance training (RT) enhances muscle mass and strength in patients with secondary sarcopenia but does not significantly impact physical function, such as gait speed [56].

Creating standardized criteria and employing global research efforts are essential for developing unified approaches to diagnosing and managing sarcopenia in CP patients. Importantly, sarcopenia is a frequently overlooked complication of CP, despite its significant impact on clinical outcomes. Studies have demonstrated that sarcopenia contributes to increased hospitalization rates and reduces survival in CP [57]. This circumstance dictates the need for timely diagnosis and also emphasizes the relevance of the use of PERT, which remains the sole effective treatment to correct EPI and prevent malnutrition [58,59,60]. The effectiveness of PERT in resolving the signs of EPI according to laboratory criteria (increased fat absorption coefficient), leveling malabsorption syndrome and improving the living standards in pancreatic patients was proved by independent meta-analyses [61]. The latest consolidated European guidelines (UEG, EPC, EDS, ESPEN, ESPGHAN, ESDO, ESPCG, 2024) for the management of patients with EPI indicate that microspheres and minimicrospheres less than 2 mm are the most effective for the treatment of this pathological condition due to better dispersion [62]. The recommended minimum doses of PERT for the initial treatment of EPI in adults are 40–50 thousand lipase per main dose (3 times a day) and half the dose (20–25 thousand) for intermediate meals. This approach minimizes the probability of energy deficiency and fat-soluble vitamins [63].

The present meta-analysis has several limitations. High heterogeneity among the included studies is a major concern, as it suggests significant differences in study design, population characteristics, and methodologies, making it challenging to draw firm conclusions. One of the primary contributors to this heterogeneity is the lack of universally accepted diagnostic criteria for sarcopenia. The use of different assessment methods (DEXA, CT, MRI, handgrip strength tests) across studies complicates direct comparisons and reduces the consistency of prevalence estimates.

Furthermore, the diverse patient populations included in the analysis, including both European and Asian cohorts, introduce potential regional variability in sarcopenia prevalence and underlying risk factors. Additionally, small sample sizes in some studies reduce the statistical power of our findings, potentially limiting the ability to detect true associations.

While statistical tests (Begg–Mazumdar and Egger’s regression tests) did not indicate significant publication bias, it cannot be entirely ruled out. Studies with negative or non-significant findings may be underrepresented in the literature, affecting the overall estimates. Another critical limitation is missing data regarding the exact number of CP patients with sarcopenia in some included studies, which weakens the completeness of the meta-analysis.

Moreover, the methodological quality assessment using the NOS revealed that only a subset of studies (*n* = 6) had high reliability. Regardless of these limitations, publication bias was excluded based on Begg–Mazumdar and Egger’s regression tests.

Despite these limitations, this meta-analysis represents the first comprehensive synthesis of sarcopenia prevalence in CP patients, highlighting its significantly increased risk compared to healthy populations. These findings broaden current knowledge and emphasize the need for further research to refine diagnostic and treatment strategies for sarcopenia in CP.

## 5. Conclusions

Our meta-analysis indicates that sarcopenia is a common complication affecting nearly 40% of CP patients. Given the adverse long-term outcomes associated with sarcopenia, it is crucial to identify this condition early in patients with CP using modern diagnostic criteria. Among the available diagnostic tools, skeletal muscle index (SMI) evaluation, based on imaging techniques such as CT or MRI, is increasingly recognized as a reliable measure of muscle mass depletion (11 out of 16 studies used this method to define sarcopenia). However, variability in diagnostic criteria across studies remains a challenge. Establishing standardized SMI cutoffs for sarcopenia in CP patients could improve consistency in diagnosis and facilitate earlier intervention. Further research is needed to validate SMI as a routine screening tool and to optimize management strategies for sarcopenia in CP.

## Figures and Tables

**Figure 1 nutrients-17-00870-f001:**
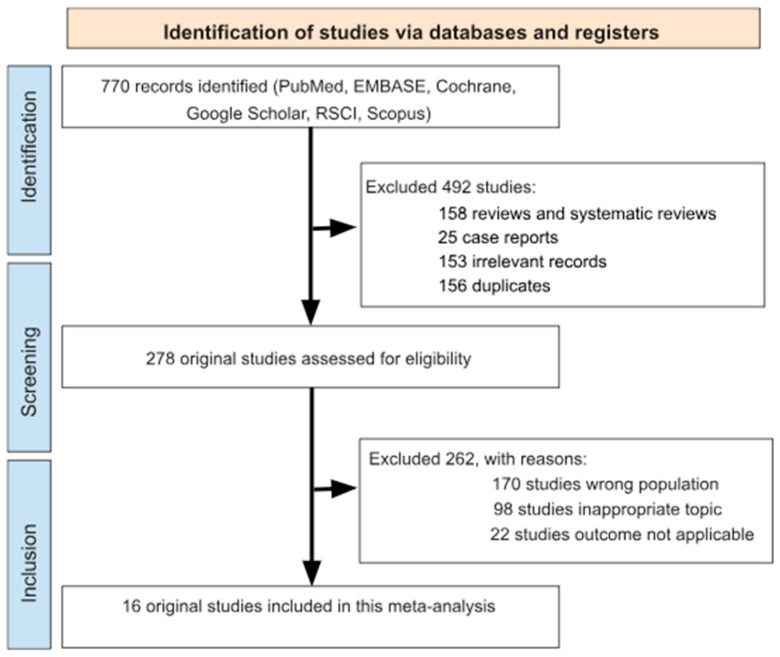
Flow chart detailing the study selection strategy.

**Figure 2 nutrients-17-00870-f002:**
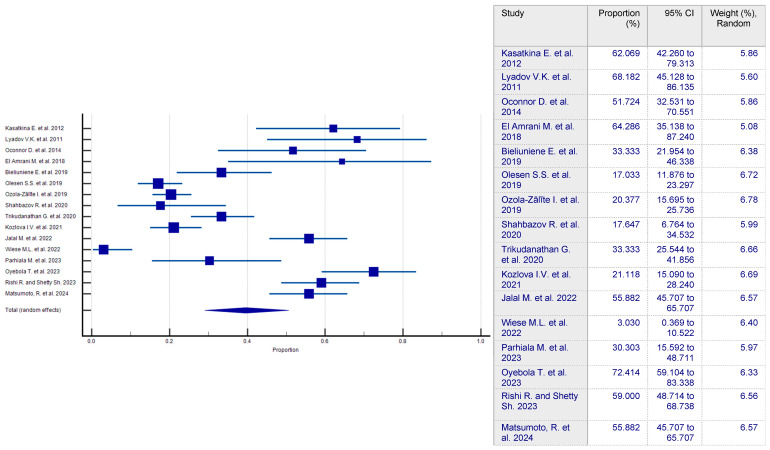
Forest plot showing the generalized prevalence of sarcopenia in CP patients [24,25,26,27,28,29,30,31,32,33,34,35,36,37,38,39].

**Figure 3 nutrients-17-00870-f003:**
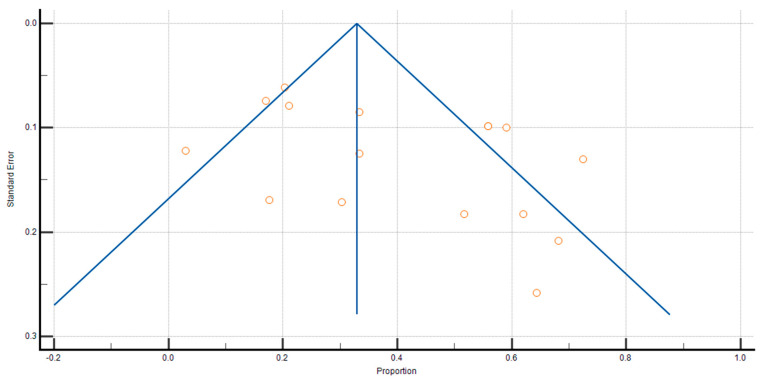
Funnel plot estimating the likelihood of publication bias when calculating the proportion of sarcopenia in patients with CP.

**Table 1 nutrients-17-00870-t001:** Characteristics of selected studies.

Authorship and Year	Country	Approach to Sarcopenia Diagnosis	Number of CP Patients	Number of Sarcopenic Patients in CP Group	Number of Patients in Control Group	Number of Sarcopenic Patients in Control Group	Evaluation of Articles According to NOS Criteria
Kasatkina E. et al., 2012 [24]	Russia	SMI measured by abdominal CT scan L3 (<52.4 cm^2^/m^2^ for men and <38.5 cm^2^/m^2^ for women)	29	18	0	0	4
Lyadov V.K. et al., 2012 [25]	Russia	SMI measured by abdominal CT scan L3 (<52.4 cm^2^/m^2^ for men and <38.5 cm^2^/m^2^ for women)	22	15	0	0	6
Oconnor D. et al., 2014 [26]	Ireland	SMI measured by abdominal CT scan L3	29	15	0	0	4
El Amrani M. et al., 2018 [27]	France	SMI measured by abdominal CT scan L3 (<52.4 cm^2^/m^2^ for men and <38.5 cm^2^/m^2^ for women)	14	9	0	0	7
Bieliuniene E. et al., 2019 [28]	Lithuania	SMI measured by abdominal CT/MRI/DEXA scan L3 (<45.4 cm^2^/m^2^ for men and <34.4 cm^2^/m^2^ for women)	63	21	0	0	6
Olesen S.S. et al., 2019 [29]	Denmark	Bioimpedance analysis with determination of the index of muscle (lean) mass (<10.76 kg/m^2^ for men; <6.76 kg/m^2^ for women)	182	31	0	0	6
Ozola-Zālīte I. et al., 2019 [30]	Lithuania	SMI measured by abdominal CT scan L4 (SMI < 41.3 cm^2^/m^2^ for men; <34.2 cm^2^/m^2^ for women)	265	54	0	0	7
Shahbazov R. et al., 2020 [31]	United States of America	CT scan to determine the thickness of the lumbar muscle (<492 mm^2^/m^2^ for men and <362 mm^2^/m^2^ for women)	34	6	0	0	7
Trikudanathan G. et al., 2020 [32]	United States of America	SMI measured by abdominal CT scan L3 (<52.4 cm^2^/m^2^ for men and <38.5 cm^2^/m^2^ for women)	138	46	0	0	6
Kozlova I.V. et al., 2021 [33]	Russia	The short physical performance battery—SPPB + handgrip strength (SPPB: ≤11 points in men and ≤10 points in women; handgrip strength: <27 kg for men and <16 kg for women)	161	34	30	0	6
Jalal M. et al., 2022 [34]	Great Britain	SMI measured by abdominal CT scan L3 (<41 cm^2^/m^2^ for women and <43 cm^2^/m^2^ if BMI 25 kg/m^2^ or 53 cm^2^/m^2^ if BMI ≥ 25 kg/m^2^ for men)	102	57	0	0	6
Wiese M.L. et al., 2022 [35]	Germany	SMI measured by abdominal CT scan L3 + handgrip strength (for men < 8.97 kg/m^2^, for women < 6.68 kg/m^2^, handgrip strength: <27 kg for men and <16 kg for women)	66	2	66	0	8
Parhiala M. et al., 2023 [36]	Finland	CT/MRI calculation of the Psoas muscle area (average area of both left and right lumbar muscles at the level of the middle of L3)—less than 800 mm^2^ for men and less than 550 mm^2^ for women	33	10	0	0	8
Oyebola T. et al., 2023 [37]	Great Britain	CT scan with determination of the psoas musculoskeletal index: total psoas muscle cross-sectional area at L3 level (cm^2^)/the patient’s height squared (m^2^)	58	42	62	28	7
Rishi R. and Shetty Sh. 2023 [38]	India	SMI measured by abdominal CT scan L3	100	59	0	0	4
Matsumoto R. et al., 2024 [39]	Japan	SMI measured by abdominal CT scan L3 (42 cm^2^/m^2^ for men and 38 cm^2^/m^2^) + handgrip strength (for men less than 28 kg and less than 18 kg for women)	102	57	0	0	6

DEXA—Dual-energy X-ray absorptiometry. SMI—Skeletal Mass Index. CT—Computed Tomography. L3—Third Lumbar Vertebra Level. L4—Fourth Lumbar Vertebra Level. SPPB—Short physical performance battery. MRI—Magnetic Resonance Imaging. PMI—Psoas muscle index.

## Data Availability

No new data were created or analyzed in this study. Data sharing is not applicable to this article.

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
