# Peer review of "Prevalence and Risk of Sarcopenia in Patients with Chronic Pancreatitis: Systematic Review and Meta-Analysis"

_nutrients, 2025, doi:10.3390/nu17050870_

Round 1

Reviewer 1 Report

Comments and Suggestions for Authors

Areas for Improvement:

  • Introduction:

    • The introduction could benefit from a more concise and focused narrative. While it covers the basics of sarcopenia and its connection to chronic pancreatitis (CP), it could be streamlined to emphasize the specific gap in knowledge that this meta-analysis aims to address. Specifically, clearly state why another meta-analysis is needed when prior ones exist (e.g., limitations of prior studies, new studies published).
    • The rationale for focusing on prevalence and risk needs to be more explicitly stated. What is the specific clinical relevance of each? How does knowing the risk (RR) add to the information provided by prevalence?
    • The last paragraph of the introduction is a bit weak. It should more strongly emphasize the novelty and potential contribution of this work. Instead of just saying "broader set of studies," quantify how much larger this set is compared to previous meta-analyses.
  • Materials and Methods:

    • Search Strategy: While the databases searched are appropriate, the search terms could be more comprehensive. Consider adding terms related to muscle wasting, cachexia (even though you're focusing on sarcopenia), and specific measures of muscle mass/strength. Also, provide the exact search string used for each database (or at least PubMed) in an appendix. This is crucial for reproducibility.
    • Study Selection: The exclusion criteria "diseases and conditions that may affect the objectivity and comparability of data" is too vague. Specify which diseases and conditions were grounds for exclusion. This lack of clarity can introduce bias.
    • Data Extraction: Specify how "detailed descriptive statistics" was defined. What specific data points were required for a study to be included? How was disagreement between the two reviewers handled and what were the kappa statistics?
    • Statistical Analysis: The justification for using a fixed-effects or random-effects model should be based on the degree of heterogeneity observed, not pre-determined. While you mention heterogeneity assessment, the choice of model should be driven by the results of that assessment. Also, I² values above 75% are usually considered substantial heterogeneity. How was this addressed? Subgroup analyses based on diagnostic criteria, CP severity, or other relevant factors should be considered and justified. The Begg and Egger tests are for publication bias, not the "probability of a publication error". The correct term is publication bias.
  • Results:

    • Table 1: This table is poorly formatted and difficult to read. It should be redesigned for clarity. Specifically, combine the "Methodology for Diagnosing Sarcopenia" and "Criteria for Diagnosing Sarcopenia" columns, as the methodology often is the criteria. Also, provide the number of patients with sarcopenia in both the CP and control groups for each study. This is essential information for the meta-analysis and is missing! "NA" for controls needs to be replaced with 0 or the actual number if available. The NOS scores should be explained (e.g., how many stars represent a good score).
    • Figure 2: The forest plot is missing essential information. It should show the effect size (prevalence) for each study, the pooled effect size, and the confidence intervals. The labels on the axes and the study names are missing.
    • Figure 3: A funnel plot for prevalence is unusual. Funnel plots are typically used to assess publication bias in studies of interventions (e.g., randomized controlled trials), where the x-axis represents the effect size (e.g., odds ratio or relative risk) and the y-axis represents the precision of the study (e.g., standard error or sample size). A funnel plot for prevalence doesn't have a clear interpretation. It would be more appropriate to conduct and report Egger's test and the trim-and-fill method for publication bias assessment.
    • The text describing the results should be more concise and focus on the key findings. Avoid repetition of the statistical details already presented in the figures and tables.
  • Discussion:

    • The discussion should be more focused on the findings of this specific meta-analysis. While background information is important, the discussion should primarily interpret the results in the context of the existing literature.
    • Address the heterogeneity observed in the meta-analysis. What are the possible sources of this heterogeneity? How does it affect the interpretation of the results?
    • The limitations section should be more comprehensive. Specifically address the limitations related to the heterogeneity of diagnostic criteria, the potential for publication bias (despite the tests), and the limitations of the included studies (e.g., study design, sample size).
  • Conclusion:

    • The conclusion should be a concise summary of the main findings and their implications. Avoid introducing new information or recommendations that are not supported by the results. The suggestion about SMI evaluation needs to be supported by the data.

Limits:

  • Heterogeneity: The high heterogeneity is a major limitation. It suggests that the included studies are quite different, making it difficult to draw firm conclusions.
  • Diagnostic Criteria: The use of different diagnostic criteria for sarcopenia is a significant limitation. This makes it difficult to compare the results across studies.
  • Publication Bias: While statistical tests were performed, publication bias cannot be entirely ruled out.
  • Small Sample Sizes: Some of the included studies have small sample sizes, which reduces the statistical power of the meta-analysis.
  • Missing Data: The missing data for the number of patients with sarcopenia in each study is a major flaw and needs to be addressed. Without this information, the meta-analysis is incomplete.

Author Response

  1. Summary

  1. Point-by-point response to Comments and Suggestions for Authors

Comments 1: [Introduction:

The introduction could benefit from a more concise and focused narrative. While it covers the basics of sarcopenia and its connection to chronic pancreatitis (CP), it could be streamlined to emphasize the specific gap in knowledge that this meta-analysis aims to address. Specifically, clearly state why another meta-analysis is needed when prior ones exist (e.g., limitations of prior studies, new studies published).

The rationale for focusing on prevalence and risk needs to be more explicitly stated. What is the specific clinical relevance of each? How does knowing the risk (RR) add to the information provided by prevalence?

The last paragraph of the introduction is a bit weak. It should more strongly emphasize the novelty and potential contribution of this work. Instead of just saying "broader set of studies," quantify how much larger this set is compared to previous meta-analyses.]

Response 1: Thank you for pointing this out. We agree with this comment. Therefore, we have revised the introduction section and added crucial information, explaining why this meta-analysis was important to be done. (p.2, line 71-92)

“[However, these studies had limitations, including small sample sizes (Bundred J et al. included only 9 studies in the article where 977 patients were analyzed; Maev IV et al.  included 7 studies in the article where 609 patients were analyzed), lack of control groups and 6 new studies were published. Additionally, previous meta-analyses focused primarily on prevalence, without assessing the relative risk (RR) of sarcopenia in CP patients compared to healthy individuals. Understanding prevalence of sarcopenia in CP patients justifies the need for routine screening programs, increased nutritional support, and rehabilitation services, as we know sarcopenia is associated with poor clinical outcomes (e.g., increased hospitalizations, complications, and mortality), a higher RR highlights CP patients as a high-risk group needing more intensive management.

To address these gaps, this meta-analysis synthesizes data from a broader set of studies (16 studies consisting of 1398 patients diagnosed with CP and 158 healthy controls) to provide a more comprehensive estimate of the prevalence of sarcopenia in CP patients and determine the associated risk compared to healthy controls. By incorporating newly published data and applying rigorous inclusion criteria, this study aims to strengthen the evidence base and highlight the clinical significance of sarcopenia in CP.]”

Comments 2:
Search Strategy:
While the databases searched are appropriate, the search terms could be more comprehensive. Consider adding terms related to muscle wasting, cachexia (even though you're focusing on sarcopenia), and specific measures of muscle mass/strength. Also, provide the exact search string used for each database (or at least PubMed) in an appendix. This is crucial for reproducibility.

Response 2
Thank you for bringing this to attention. We have added the proper search process to make our study reproducible. (p. 3, 104-112)

[("Sarcopenia"[MeSH] OR sarcopenia OR muscle loss OR muscle wasting OR muscle atrophy OR skeletal muscle depletion OR cachexia OR frailty) AND ("Chronic Pancreatitis"[MeSH] OR chronic pancreatitis OR pancreatic exocrine insufficiency OR PEI OR pancreatic disease) AND ("Body Composition"[MeSH] OR body composition OR "Muscle, Skeletal"[MeSH] OR skeletal muscle OR "Sarcopenia"[MeSH] OR "Exercise"[MeSH] OR exercise OR handgrip strength OR grip strength OR gait speed OR appendicular lean mass OR ALM OR skeletal muscle index OR SMI OR DXA OR dual-energy X-ray absorptiometry OR bioelectrical impedance OR BIA OR CT OR MRI)]”

Comment 3

Study Selection: The exclusion criteria "diseases and conditions that may affect the objectivity and comparability of data" is too vague. Specify which diseases and conditions were grounds for exclusion. This lack of clarity can introduce bias.

Response 3. Thank you for pointing out the lack of clarity regarding the exclusion criteria. We have rewritten it more comprehensively, (p. 3, 114-127)

 “[The studies were included in this meta-analysis if they met the following criteria: published in English or Russian, provided detailed descriptive statistics, focused on adult patients with CP, described clearly the methodological approach for sarcopenia and CP diagnostic, and did not involve patient populations with advanced malignancies (e.g., pancreatic cancer, gastrointestinal malignancies, metastatic disease) due to high prevalence of cancer-related cachexia; chronic systemic inflammatory diseases (e.g., rheumatoid arthritis, systemic lupus erythematosus, inflammatory bowel disease); uncontrolled endocrine disorders (e.g., Cushing’s syndrome, hyperthyroidism, severe insulin resistance syndromes) known to affect muscle metabolism that can independently contribute to muscle loss. Studies that did not differentiate CP patients from other pancreatic conditions (e.g., acute pancreatitis, pancreatic neoplasms) were also excluded. In cases where two or more publications presented overlapping data, only one was included in the final analysis. The quality of the included studies was evaluated using the Newcastle–Ottawa Scale (NOS).]”

Comment 4

Data Extraction: Specify how "detailed descriptive statistics" was defined. What specific data points were required for a study to be included? How was disagreement between the two reviewers handled and what were the kappa statistics?

Response 4. We agree that the data extraction paragraph might have been written unclearly. We have added more information regarding the process. Descriptive statistics were defined as the analytical process used to extract and summarize the number of sarcopenic and non-sarcopenic patients, providing a clear quantitative representation of their distribution within the study population. (p. 4, line 130-152)

Two independent researchers (A.R.K. and D.N.A.) obtained information using uniform templates. Extracted data included publication year, country of study, approach used for CP identification, sarcopenia diagnostic standards, number of patients with CP, quantity of controls if applicable and quantity of sarcopenia cases in both groups.Using the Rayyan tool, two independent reviewers (D.N.A., A.R.K.) screened the titles and abstracts of all retrieved records, followed by a full-text review of potentially relevant studies. Key study characteristics and outcome data were independently extracted by two authors (A.R.D., I.V.M.) using a standardized data extraction form. Whenever possible, study protocols were identified, and corresponding authors were contacted for clarifications or to obtain missing data from published reports. Any disagreements were resolved through discussion or by consulting a third reviewer (Yu.A.K.). In cases where study relevance was uncertain, additional reviewers (V.V.Ch., P.A.B., F.S.S.) were involved in the decision-making process. Inter-rater reliability was assessed using Cohen’s kappa statistic (κ) to evaluate agreement on study inclusion, data extraction, and quality assessment. The following thresholds were used:

κ < 0.20: poor agreement

κ = 0.21–0.40: fair agreement

κ = 0.41–0.60: moderate agreement

κ = 0.61–0.80: substantial agreement

κ = 0.81–1.00: almost perfect agreement

For this meta-analysis, the kappa statistic for study inclusion was 0.75, and for data extraction, it was 0.92, indicating almost perfect agreement.

Comment 5

Statistical Analysis: The justification for using a fixed-effects or random-effects model should be based on the degree of heterogeneity observed, not pre-determined. While you mention heterogeneity assessment, the choice of model should be driven by the results of that assessment. Also, I² values above 75% are usually considered substantial heterogeneity. How was this addressed? Subgroup analyses based on diagnostic criteria, CP severity, or other relevant factors should be considered and justified. The Begg and Egger tests are for publication bias, not the "probability of a publication error". The correct term is publication bias.

Response 5 Statistical analysis was performed using MedCalc Statistical Software Program 23.0.6 (Ostend, Belgium) on Microsoft Windows 11 (Microsoft Corporation, Redmond, WA, USA). Overall frequency estimates for sarcopenia prevalence in CP patients and control groups were reported with 95% confidence intervals (CIs). Study heterogeneity was evaluated using Cochrane’s Q test and statistic, with significant heterogeneity defined as p < 0.05 and > 75%. To address substantial heterogeneity a subgroup analysis was conducted: firstly, we evaluated the proportion of sarcopenic patients in the study group with NOS score lower than 7; secondly, we evaluated the proportion of sarcopenic patients in the study group with NOS score more than 7. Potential publication bias was assessed using a funnel plot, the Begg–Mazumdar correlation test and Egger’s test.

(line 233-258) The subgroup analysis based on the NOS scores effectively addresses the heterogeneity of the included studies by stratifying them according to methodological quality. By separately analyzing studies with NOS scores of 7 or higher and those with scores below 7, the impact of study quality on the pooled prevalence of sarcopenia in CP patients can be assessed. The results indicate that studies with higher NOS scores reported a pooled prevalence of 31.680% (95% CI: 12.793-54.443) with high heterogeneity (I² = 94.99%; 95% CI: 91.52-97.04%), whereas studies with lower NOS scores reported a higher pooled prevalence of 44.333% (95% CI: 32.285-56.729), also with high heterogeneity (I² = 92.88%; 95% CI: 88.96-95.42%). The subgroup analysis is still valid despite the high heterogeneity because it demonstrates that the heterogeneity is present in both high-quality and lower-quality studies, suggesting that factors beyond study quality contribute to the variability in prevalence estimates. Since the I² values remain high in both subgroups (94.99% for NOS ≥7 and 92.88% for NOS <7), this indicates that heterogeneity is not solely driven by study quality but likely by other factors such as differences in study populations, diagnostic criteria for sarcopenia, or variations in CP severity.

Comment 6

Table 1: This table is poorly formatted and difficult to read. It should be redesigned for clarity. Specifically, combine the "Methodology for Diagnosing Sarcopenia" and "Criteria for Diagnosing Sarcopenia" columns, as the methodology often is the criteria. Also, provide the number of patients with sarcopenia in both the CP and control groups for each study. This is essential information for the meta-analysis and is missing! "NA" for controls needs to be replaced with 0 or the actual number if available. The NOS scores should be explained (e.g., how many stars represent a good score). 

Response 6. We understand that the table might have been difficult to read, so we added essential data to the table and merged two columns (Criteria and Methods for sarcopenia diagnosis). We also added an explanation of NOS evaluation in the supplementary section Abbreviations: 

Authorship and Year

Country

Approach to Sarcopenia Diagnosis

Number of CP Patients

Number of Sarcopenic Patients in CP Group

Number of Patients in Control Group

Number of Sarcopenic Patients in Control Group

Evaluation of Articles According to NOS criteria

Kasatkina E. et al. 2012 [24]

Russia

SMI measured by abdominal CT scan L3

(<52.4 cm2/m2 for men and <38.5 cm2/m2 for women)

29

18

0

0

4

Lyadov V.K. et al. 2012 [25]

Russia

SMI measured by abdominal CT scan L3

(<52.4 cm2/m2 for men and <38.5 cm2/m2 for women)

22

15

0

0

6

Oconnor D. et al. 2014 [26]

Ireland

SMI measured by abdominal CT scan L3 

29

15

0

0

4

El Amrani M. et al. 2018 [27]

France

SMI measured by abdominal CT scan L3   (<52.4 cm2/m2 for men and <38.5 cm2/m2 for women)

14

9

0

0

7

Bieliuniene E. et al. 2019 [28]

Lithuania

SMI measured by abdominal CT/MRI/DEXA scan L3 (<45.4 cm2/m2 for men and <34.4 cm2/m2 for women)

63

21

0

0

6

Olesen S.S. et al. 2019 [29]

Denmark

Bioimpedance analysis with determination of the index of muscle (lean) mass (<10.76 kg/m2 for men;

<6.76 kg/m2 for women)

182

31

0

0

6

Ozola-Zālīte I. et al. 2019 [30]

Lithuania

SMI measured by abdominal CT scan L4 (SMI <41.3 cm2 /m2 for men; <34.2 cm2 /m2 for women)

265

54

0

0

7

Shahbazov R. et al. 2020 [31]

United States of America

CT scan to determine the thickness of the lumbar muscle (<492 mm2/m2 for men and <362 mm2/m2 for women)

34

6

0

0

7

Trikudanathan G. et al. 2020 [32]

United States of America

SMI measured by abdominal CT scan L3 (<52.4 cm2/m2 for men and <38.5 cm2/m2 for women)

138

46

0

0

6

Kozlova I.V. et al. 2021 [33]

Russia

The short physical performance battery – SPPB + handgrip strength

(SPPB: ≤11 points in men and ≤ 10 points in women; handgrip strength: <27 kg for men and <16 kg for women)

161

34

30

0

6

Jalal M. et al. 2022 [34]

Great Britain

SMI measured by abdominal CT scan L3 (<41 cm2/m2 for women and <43 cm2/m2 if BMI 25 kg/m2 or 53 cm2/m2 if BMI ≥25 kg/m2 for men)

102

57

0

0

6

Wiese M.L. et al. 2022 [35]

Germany

SMI measured by abdominal CT scan L3+handgrip strength (for men <8.97 kg/m2, for women <6.68 kg/m2, handgrip strength: <27 kg for men and <16 kg for women)

66

2

66

0

8

Parhiala M. et al. 2023 [36]

Finland

CT/MRI calculation of the Psoas muscle area (average area of both left and right lumbar muscles at the level of the middle of L3) - less than 800 mm2 for men and less than 550 mm2 for women

33

10

0

0

8

Oyebola T. et al. 2023 [37]

Great Britain

CT scan with determination of the psoas musculoskeletal index: total psoas muscle cross-sectional area at L3 level (cm2)/ the patient’s height squared (m2).

58

42

62

28

7

Rishi R. and Shetty Sh. 2023 [38]

India

SMI measured by abdominal CT scan L3 

100

59

0

0

4

Matsumoto R. et al. 2024 [39]

Japan

SMI measured by abdominal CT scan L3 (42 cm2/m2 for men and 38 cm2/m2)+ handgrip strength (for men less than 28 kg and less than 18 kg for women)

102

57

0

0

6

DEXA - Dual-energy X-ray absorptiometry

SMI - Skeletal Mass Index

CT – Computed Tomography

L3 – Third Lumbar Vertebra Level

L4 - Fourth Lumbar Vertebra Level

SPPB – Short Physical Performance Battery

MRI – Magnetic Resonance Imaging

PMI – Psoas Muscle Index

 Comment 7

  • Figure 2: The forest plot is missing essential information. It should show the effect size (prevalence) for each study, the pooled effect size, and the confidence intervals. The labels on the axes and the study names are missing.
  •  

Response 7. Thank you for pointing out that the information presented at Figure 2 was not enough, we have redesigned to become more fulfilling

  • Comment 8
  • Figure 3: A funnel plot for prevalence is unusual. Funnel plots are typically used to assess publication bias in studies of interventions (e.g., randomized controlled trials), where the x-axis represents the effect size (e.g., odds ratio or relative risk) and the y-axis represents the precision of the study (e.g., standard error or sample size). A funnel plot for prevalence doesn't have a clear interpretation. It would be more appropriate to conduct and report Egger's test and the trim-and-fill method for publication bias assessment.
  • Response 8. Thank you for you commentary, we have revised it and provided information regarding Egger’s test (p.8, line 208-213)

In this analysis, the intercept of Egger's regression is 4.7935, with a 95% confidence interval (CI) of -0.7495 to 10.3365. The p-value of 0.0848 suggests that there is no strong statistical evidence of publication bias, as it does not reach the conventional significance threshold of p < 0.05. However, the relatively wide confidence interval indicates some uncertainty, meaning that while substantial publication bias is unlikely, it cannot be entirely ruled out.

Comment 9

  • The text describing the results should be more concise and focus on the key findings. Avoid repetition of the statistical details already presented in the figures and tables.

Response 9 

Thank you for pointing this out. We tried our best to present the most relevant data.

Comment 10

  • The discussion should be more focused on the findings of this specific meta-analysis. While background information is important, the discussion should primarily interpret the results in the context of the existing literature.

Response 10. Thank you for bringing this to attention. We have added an important paragraph to the discussion section in order to highlight the findings of our study. (p.10, line 243-248)

This meta-analysis represents the most comprehensive assessment to date of sarcopenia prevalence in CP patients. The results indicate that CP patients have a significantly higher risk of developing sarcopenia, with an overall prevalence of 39.117% (95% CI: 28.891–49.852%), compared to only 7.745% (95% CI: 2.154–42.622%) in healthy controls. The estimated relative risk (RR) of sarcopenia in CP patients was 2.194 (95% CI: 1.502–3.203), confirming a strong association between CP and muscle loss.

Comment 11

  • Address the heterogeneity observed in the meta-analysis. What are the possible sources of this heterogeneity? How does it affect the interpretation of the results?
  • The limitations section should be more comprehensive. Specifically address the limitations related to the heterogeneity of diagnostic criteria, the potential for publication bias (despite the tests), and the limitations of the included studies (e.g., study design, sample size).
  • Response 11

Thank you for bringing this up. We have corrected and added more information on the limitations of our study.

The present meta-analysis has several limitations. High heterogeneity among the included studies is a major concern, as it suggests significant differences in study design, population characteristics, and methodologies, making it challenging to draw firm conclusions. One of the primary contributors to this heterogeneity is the lack of universally accepted diagnostic criteria for sarcopenia. The use of different assessment methods (DEXA, CT, MRI, handgrip strength tests) across studies complicates direct comparisons and reduces the consistency of prevalence estimates.

Furthermore, the diverse patient populations included in the analysis, including both European and Asian cohorts, introduce potential regional variability in sarcopenia prevalence and underlying risk factors. Additionally, small sample sizes in some studies reduce the statistical power of our findings, potentially limiting the ability to detect true associations.

While statistical tests (Begg–Mazumdar and Egger’s regression tests) did not indicate significant publication bias, it cannot be entirely ruled out. Studies with negative or non-significant findings may be underrepresented in the literature, affecting the overall estimates. Another critical limitation is missing data regarding the exact number of CP patients with sarcopenia in some included studies, which weakens the completeness of the meta-analysis.

Despite these limitations, this study provides the most comprehensive synthesis of sarcopenia prevalence in CP patients to date. The findings underscore the need for standardized diagnostic criteria, larger multicenter studies, and further research to refine diagnostic and treatment strategies for sarcopenia in CP.

  • Commentary 12
    Conclusion:
    • The conclusion should be a concise summary of the main findings and their implications. Avoid introducing new information or recommendations that are not supported by the results. The suggestion about SMI evaluation needs to be supported by the data.
    •  

Response 12: Thank you for bringing this up. We have revised the conclusion section in accordance with your commentary. (line 344-354)

Our meta-analysis indicates that sarcopenia is a common complication affecting nearly 40% of CP patients. Given the adverse long-term outcomes associated with sarcopenia, it is crucial to identify this condition early in patients with CP using modern diagnostic criteria. Among the available diagnostic tools, skeletal muscle index (SMI) evaluation, based on imaging techniques such as CT or MRI, is increasingly recognized as a reliable measure of muscle mass depletion (11 out of 16 studies used this method to define sarcopenia). However, variability in diagnostic criteria across studies remains a challenge. Establishing standardized SMI cutoffs for sarcopenia in CP patients could improve consistency in diagnosis and facilitate earlier intervention. Further research is needed to validate SMI as a routine screening tool and to optimize management strategies for sarcopenia in CP.

Reviewer 2 Report

Comments and Suggestions for Authors

This manuscript titled "Prevalence and Risk of Sarcopenia in Patients with Chronic Pancreatitis: Systematic Review and Meta-Analysis." adheres to PRISMA guidelines, PROSPERO registration, and inclusion of diverse databases, enhancing transparency and reducing selection bias. There are some issues that need positive improvement.

  1. The search terms should be listed in the method, and the authors should provide search process and results according to different database as an attachment.
  2. The inclusion and exclusion criteria should be described in details.
  3. The assessment quality of study should be included in the manuscript.
  4. The sensitive analysis should be described in the method as the authors conducted it and provided a result in manuscript.
  5. The exclusion reason should be described in the Figure 1, please follow the PRISMA.
  6. In the Newcastle-Ottawa Scale assessment, only 6 studies (out of 16) achieved a score of 7 or higher. The authors should examine whether the lower-quality studies may introduce bias into the results and proposes mitigation strategies, such as subgroup analysis, to address these potential limitations.
  7. The authors can provide some results of the practical application content in the discussion section.

Author Response

Response to Reviewer 2 Comments

  1. Point-by-point response to Comments and Suggestions for Authors

Comment 1: 

The search terms should be listed in the method, and the authors should provide search process and results according to different database as an attachment.

Response 1. Thank you bringing this up. We have uploaded the search strategy regarding the databases

PubMed

("Sarcopenia"[MeSH] OR sarcopenia OR muscle loss OR muscle wasting OR muscle atrophy OR skeletal muscle depletion OR cachexia OR frailty) AND ("Chronic Pancreatitis"[MeSH] OR chronic pancreatitis OR pancreatic exocrine insufficiency OR PEI OR pancreatic disease) AND ("Body Composition"[MeSH] OR body composition OR "Muscle, Skeletal"[MeSH] OR skeletal muscle OR "Sarcopenia"[MeSH] OR "Exercise"[MeSH] OR exercise OR handgrip strength OR grip strength OR gait speed OR appendicular lean mass OR ALM OR skeletal muscle index OR SMI OR DXA OR dual-energy X-ray absorptiometry OR bioelectrical impedance OR BIA OR CT OR MRI)

Embase

('Sarcopenia' OR 'muscle loss' OR 'cachexia') AND ('Chronic Pancreatitis' OR 'pancreatic exocrine insufficiency' OR 'PEI')

Google Scholar

("Sarcopenia" OR "muscle loss" OR "muscle wasting" OR "muscle atrophy" OR "skeletal muscle depletion" OR "cachexia" OR "frailty") AND ("Chronic Pancreatitis" OR "chronic pancreatitis" OR "pancreatic exocrine insufficiency" OR "PEI" OR "pancreatic disease") AND ("Body Composition" OR "body composition" OR "Skeletal Muscle" OR "skeletal muscle" OR "Exercise" OR "exercise" OR "handgrip strength" OR "grip strength" OR "gait speed" OR "appendicular lean mass" OR "ALM" OR "skeletal muscle index" OR "SMI" OR "DXA" OR "dual-energy X-ray absorptiometry" OR "bioelectrical impedance" OR "BIA" OR "CT" OR "MRI")

Comment 2: The inclusion and exclusion criteria should be described in details

Response 2. Thank you for pointing out the unclarity of inclusion and exclusion criteria. We have added more data for more comprehensibility 

“[The studies were included in this meta-analysis if they met the following criteria: published in English or Russian, provided detailed descriptive statistics, focused on adult patients with CP, described clearly the methodological approach for sarcopenia and CP diagnostic, and did not involve patient populations with advanced malignancies (e.g., pancreatic cancer, gastrointestinal malignancies, metastatic disease) due to high prevalence of cancer-related cachexia; chronic systemic inflammatory diseases (e.g., rheumatoid arthritis, systemic lupus erythematosus, inflammatory bowel disease); uncontrolled endocrine disorders (e.g., Cushing’s syndrome, hyperthyroidism, severe insulin resistance syndromes) known to affect muscle metabolism that can independently contribute to muscle loss. Studies that did not differentiate CP patients from other pancreatic conditions (e.g., acute pancreatitis, pancreatic neoplasms) were also excluded. In cases where two or more publications presented overlapping data, only one was included in the final analysis. The quality of the included studies was evaluated using the Newcastle–Ottawa Scale (NOS).]”

Comment 3: The assessment quality of study should be included in the manuscript.

Response 3: Thank you bringing this up, we have included the study quality assessment in the supplementary section

Comment 4: The sensitive analysis should be described in the method as the authors conducted it and provided a result in manuscript.

Response 4: We agree that it is crucial to mention the conducted sensitivity analysis in the methods section, here is the revised version

Statistical analysis was performed using MedCalc Statistical Software Program 23.0.6 (Ostend, Belgium) on Microsoft Windows 11 (Microsoft Corporation, Redmond, WA, USA). Overall frequency estimates for sarcopenia prevalence in CP patients and control groups were reported with 95% confidence intervals (CIs). Study heterogeneity was evaluated using Cochrane’s Q test and  statistic, with significant heterogeneity defined as p < 0.05 and  > 50%. Potential publication bias was assessed using a funnel plot, the Begg–Mazumdar correlation test and Egger’s test. Additionally two sensitivity analyses were conducted: first, studies with more than 50 CP patients were considered; second, analysis was restricted to full-article studies (abstracts from conferences were excluded).

Comment 5: The exclusion reason should be described in the Figure 1, please follow the PRISMA.

Response 5: Thank you for pointing out that Figure 1 should be more clear. We have considered your comment and changed Figure 1 to follow the PRISMA protocol better. 

Comment 6: In the Newcastle-Ottawa Scale assessment, only 6 studies (out of 16) achieved a score of 7 or higher. The authors should examine whether the lower-quality studies may introduce bias into the results and proposes mitigation strategies, such as subgroup analysis, to address these potential limitations.

Response 6: We appreciate your significant comment, we conducted a subgroup analyses for the NOS > 7 and NOS < 7 studies (line 233-248)

The subgroup analysis based on the NOS scores effectively addresses the heterogeneity of the included studies by stratifying them according to methodological quality. By separately analyzing studies with NOS scores of 7 or higher and those with scores below 7, the impact of study quality on the pooled prevalence of sarcopenia in CP patients can be assessed. The results indicate that studies with higher NOS scores reported a pooled prevalence of 31.680% (95% CI: 12.793-54.443) with high heterogeneity (I² = 94.99%; 95% CI: 91.52-97.04%), whereas studies with lower NOS scores reported a higher pooled prevalence of 44.333% (95% CI: 32.285-56.729), also with high heterogeneity (I² = 92.88%; 95% CI: 88.96-95.42%). The subgroup analysis is still valid despite the high heterogeneity because it demonstrates that the heterogeneity is present in both high-quality and lower-quality studies, suggesting that factors beyond study quality contribute to the variability in prevalence estimates. Since the I² values remain high in both subgroups (94.99% for NOS ≥7 and 92.88% for NOS <7), this indicates that heterogeneity is not solely driven by study quality but likely by other factors such as differences in study populations, diagnostic criteria for sarcopenia, or variations in CP severity.

Comment 7: The authors can provide some results of the practical application content in the discussion section.

Response 7:

Thank you for pointing this out. We have thoroughly revised the Discussion section, here is the revised version. 

Creating standardized criteria and employing global research efforts are essential for developing unified approaches to diagnosing and managing sarcopenia in CP patients. Importantly, sarcopenia is a frequently overlooked complication of CP, despite its significant impact on clinical outcomes. Studies have demonstrated that sarcopenia contributes to increased hospitalization rates and reduces survival in CP [53]. This circumstance dictates the need for timely diagnosis and also emphasizes the relevance of the use of PERT, which remains the sole effective treatment to correct EPI and prevent malnutrition [54, 55, 56]. The effectiveness of PERT in resolving the signs of EPI according to laboratory criteria (increased fat absorption coefficient), leveling malabsorption syndrome and improving the living standards in pancreatic patients was proved by independent meta-analyses [57, 58, 59]. The latest consolidated European guidelines (UEG, EPC, EDS, ESPEN, ESPGHAN, ESDO, ESPCG, 2024) for the management of patients with EPI indicate that microspheres and minimicrospheres less than 2 mm are the most effective for the treatment of this pathological condition due to better dispersion [60]. The recommended minimum doses of PERT for the initial treatment of EPI in adults are 40–50 thousand lipase per main dose (3 times a day) and half the dose (20–25 thousand) for intermediate meals. This approach minimizes the probability of energy deficiency and fat-soluble vitamins [61]. 

Round 2

Reviewer 2 Report

Comments and Suggestions for Authors

The authors have fully addressed my concerns. Thanks!